# Neuro-Symbolic Entropy Regularization

**Kareem Ahmed**[1]     **Eric Wang**[1]     **Kai-Wei Chang**[1]     **Guy Van den Broeck**[1]

[1]Computer Science Department, University of California Los Angeles, USA

## Abstract

In structured output prediction, the goal is to jointly predict several output variables that together encode a structured object – a path in a graph, an entity-relation triple, or an ordering of objects. Such a large output space makes learning hard and requires vast amounts of labeled data. Different approaches leverage alternate sources of supervision. One approach – entropy regularization – posits that decision boundaries should lie in low-probability regions. It extracts supervision from unlabeled examples, but remains agnostic to the structure of the output space. Conversely, neuro-symbolic approaches exploit the knowledge that not every prediction corresponds to a *valid* structure in the output space. Yet, they do not further restrict the learned output distribution. This paper introduces a framework that unifies both approaches. We propose a loss, *neuro-symbolic entropy regularization*, that encourages the model to confidently predict a valid object. It is obtained by restricting entropy regularization to the distribution over only the valid structures. This loss can be computed efficiently when the output constraint is expressed as a tractable logic circuit. Moreover, it seamlessly integrates with other neuro-symbolic losses that eliminate invalid predictions. We demonstrate the efficacy of our approach on a series of semi-supervised and fully-supervised structured-prediction experiments, where it leads to models whose predictions are more accurate as well as more likely to be valid.

## 1 INTRODUCTION

Neural networks have achieved breakthroughs across a wide range of domains. Such breakthroughs are often only possible in the presence of large labeled datasets, which can be hard to obtain. Increasing efforts are therefore being devoted to approaches that utilize alternate sources of supervision in lieu of *more* labeled data. Entropy regularization constitutes one such approach [Grandvalet and Bengio, 2005, Chapelle et al., 2010]. It posits that data belonging to the same class tend to form discrete clusters. Minimizing the entropy of the predictive distribution can thus be regarded as minimizing a measure of class overlap under the learned representation. Intuitively, a classifier guessing uniformly at random has *maximum entropy* and has not learned features that are informative of the underlying class. Consequently, we prefer a *minimum entropy* classifier that learns features *maximally informative* of the underlying class, even on unlabeled data.

The need for labeled data is only exacerbated in structured prediction, where the objective is to predict multiple inter-dependent output variables representing a discrete object. Viewed as traditional classification, the number of classes in structured prediction is exponential in the number of output variables – all possible output configurations. Neuro-symbolic methods can provide additional supervision, leveraging symbolic knowledge regarding the structure of the output space [De Raedt et al., 2020]. This knowledge, typically expressed in logic, characterizes the set of valid structures; for instance, a path in a graph is a series of *connected* edges commencing at the source and terminating at the destination.

In this paper, we take a principled approach to unifying the aforementioned forms of supervision. Naively, we might consider simply optimizing both losses simultaneously. However, computed in that manner, entropy regularization does not account for the structure of the output space and is therefore likely to push the network towards invalid structures. Instead, we restrict the entropy loss to the network's distribution over the valid structures, as characterized by the constraint, as opposed to the entire predictive distribution, proposing *neuro-symbolic entropy regularization*. That is, we require that the network's output distribution be maximally informative of the target *subject to the constraint*. Intuitively, the network should "know" the right structure among the valid structures. Computing the entropy of a

*Accepted for the 38th Conference on Uncertainty in Artificial Intelligence* (UAI 2022).

distribution subject to a constraint is, in general, computationally hard. We provide an algorithm leveraging structural properties of tractable logical circuits to efficiently compute this quantity. Our framework integrates seamlessly with other neuro-symbolic approaches that maximize the constraint probability, in effect "eliminating" invalid structures.

Empirically, we evaluate our loss on four structured prediction tasks, in both semi-supervised and fully-supervised settings. We observe it leads to models whose predictions are more accurate, and more likely to satisfy the constraint.

**Organization** This paper is structured as follows. We start by introducing the notation and background assumed throughout the paper. Section 2 motivates, and formally defines, our neuro-symbolic entropy loss. Section 3 derives an algorithm that exploits certain structural properties of logical circuits that enable the efficient computation of our loss. Section 4 illustrates our algorithm on a toy constraint, where the probability and neuro-symbolic entropy computations are made explicit. Section 5 empirically validates our proposed approach on tasks in both semi-supervised and fully-supervised settings. Section 6 reviews, and draws connections to the the neuro-symbolic and the semi-supervised literatures. We step through an example compiling a logical formula in Section A and conclude in Section 7. Our code can be found at `https://github.com/UCLA-StarAI/NeSyEntropy`.

# 2 NEURO-SYMBOLIC ENTROPY LOSS

We first introduce background on logical constraints and probability distributions over output structures. Afterwards, we motivate and define our neuro-symbolic entropy loss.

## 2.1 BACKGROUND

We write uppercase letters $(X, Y)$ for Boolean variables and lowercase letters $(x, y)$ for their instantiation ($Y = 0$ or $Y = 1$). Sets of variables are written in bold uppercase $(\mathbf{X}, \mathbf{Y})$, and their joint instantiation in bold lowercase $(\mathbf{x}, \mathbf{y})$. A literal is a variable $(Y)$ or its negation $(\neg Y)$. A logical sentence ($\alpha$ or $\beta$) is constructed from variables and logical connectives ($\wedge, \vee$, etc.), and is also called a (logical) formula or constraint. A state or world $\mathbf{y}$ is an instantiation to all variables $\mathbf{Y}$. A state $\mathbf{y}$ satisfies a sentence $\alpha$, denoted $\mathbf{y} \models \alpha$, if the sentence evaluates to true in that world. A state $\mathbf{y}$ that satisfies a sentence $\alpha$ is also said to be a model of $\alpha$. We denote by $m(\alpha)$ the set of all models of $\alpha$. The notation for states $\mathbf{y}$ is used to refer to an assignment, the logical sentence enforcing the assignment, or the binary output vector capturing the assignment, as these are all equivalent notions. A sentence $\alpha$ entails another sentence $\beta$, denoted $\alpha \models \beta$, if all worlds that satisfy $\alpha$ also satisfy $\beta$.

**A Probability Distribution over Possible Structures** Let $\alpha$ be a logical sentence defined over Boolean variables $\mathbf{Y} = \{Y_1, \ldots, Y_n\}$. Let $\mathbf{p}$ be a vector of probabilities for the same variables $\mathbf{Y}$, where $\mathbf{p}_i$ denotes the predicted probability of variable $Y_i$ and corresponds to a single output of the neural network. The neural network's outputs induce a probability distribution $P(\cdot)$ over all possible states $\mathbf{y}$ of $\mathbf{Y}$:

$$P(\mathbf{y}) = \prod_{i:\mathbf{y}\models\mathbf{Y}_i} \mathbf{p}_i \prod_{i:\mathbf{y}\models\neg\mathbf{Y}_i} (1 - \mathbf{p}_i). \qquad (1)$$

**Semantic Loss** The semantic loss [Xu et al., 2018] is a function of the logical constraint $\alpha$ and a probability vector $\mathbf{p}$. It quantifies how close the neural network comes to satisfying the constraint by computing the probability of the constraint under the distribution $P(\cdot)$ induced by $\mathbf{p}$. It does so by reducing the problem of probability computation to weighted model counting (WMC): summing up the models of $\alpha$, each weighted by its likelihood under $P(\cdot)$. It, therefore, maximizes the probability mass allocated by the network to the models of $\alpha$

$$\mathbb{E}_{\mathbf{y}\sim P}\left[\mathbb{1}\{\mathbf{y} \models \alpha\}\right] = \sum_{\mathbf{y}\models\alpha} P(\mathbf{y}). \qquad (2)$$

Taking the negative logarithm recovers semantic loss. We make use of semantic loss in our experiments to "eliminate" invalid structures under the neural network's distribution.

## 2.2 MOTIVATION AND DEFINITION

Consider the plots in Figure 1. For any given data point $x$, the neural network can be fairly uncertain regarding the target class, accommodating for both valid and invalid structured predictions under its predicted distribution.

A common underlying assumption in many machine learning methods is that data belonging to the same class tend to form discrete clusters [Chapelle et al., 2010] – an assumption deemed justified on the sheer basis of the existence of classes. Consequently, a classifier is expected to favor decision boundaries lying in regions of low data density, separating the clusters. Entropy-regularization [Grandvalet and Bengio, 2005] directly implements the above assumption, requiring that the classifier output confident – low-entropy – predictive distributions, pushing the decision boundary away from unlabeled points, thereby supplementing scarce labeled data with abundant unlabeled data. Seen through that lens, minimizing the entropy of the predictive distribution can be regarded as minimizing a measure of class overlap as a function of the features learned by the network.

Entropy regularization, however, remains agnostic to the underlying domain, failing to exploit situations where we have knowledge characterizing valid predictions in the domain. Therefore, it can often be detrimental to a model's performance, causing it to grow confident in invalid predictions.

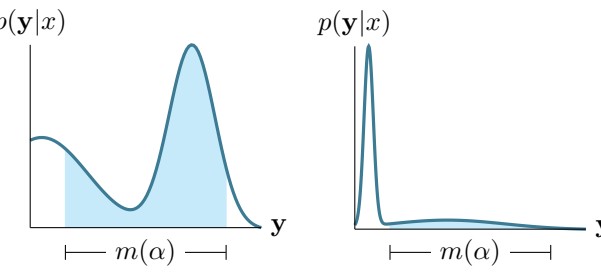

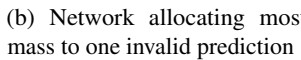

(a) Network uncertain over both valid and invalid predictions

(b) Network allocating most mass to one invalid prediction

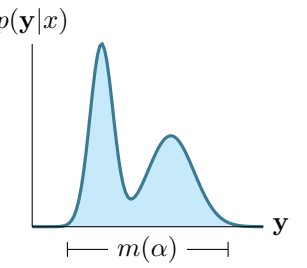

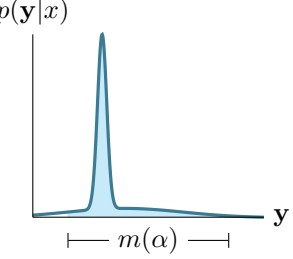

(c) Network allocating most mass to valid predictions

(d) Network allocating most mass to one valid prediction

Figure 1: A network's predictive distribution can be uncertain or certain ($\leftrightarrow$), and it can allow or disallow invalid predictions under the constraint $\alpha$ ($\updownarrow$). Entropy regularization steers the network towards confident, possibly invalid predictions (b). Neuro-symbolic learning steers the network towards valid predictions without necessarily being confident (c). Neuro-symbolic entropy-regularization guides the network to valid and confident predictions (d).

Conversely, neuro-symbolic approaches steer the network towards distributions disallowing invalid predictions, by maximizing the constraint probability, but do little to ensure the network learn features conducive to classification.

Clearly then, there is a benefit to combining the merits of both approaches. We restrict the entropy computation to the distribution over models of the logical formula, ensuring the network only grow confident in valid predictions. Complemented with maximizing the constraint probability, the network learns to allocate all of its mass to models of the constraint, while being maximally informative of the target.

**Defining the Loss**   More precisely, let $\mathbf{Y}$ be a random variable distributed according to Equation 1: $\mathbf{Y} \sim \mathrm{P}$. We are interested in minimizing the entropy of $\mathbf{Y}$ conditioned on the constraint $\alpha$

$$H(\mathbf{Y}|\alpha) = -\sum_{\mathbf{y} \models \alpha} \mathrm{P}(\mathbf{y}|\alpha) \log \mathrm{P}(\mathbf{y}|\alpha)$$

$$= -\mathbb{E}_{\mathbf{Y}|\alpha}\left[\log \mathrm{P}(\mathbf{Y}|\alpha)\right]. \tag{3}$$

---

**Algorithm 1** $\mathrm{ENT}(\alpha, \mathrm{P}, \mathsf{c})$

**Input:** a smooth, deterministic and decomposable logical circuit $\alpha$, a fully-factorized probability distribution $\mathrm{P}(\cdot)$ over states of $\alpha$, and a cache $\mathsf{c}$ for memoization

**Output:** $H(\mathbf{Y}|\alpha)$, where $\mathbf{Y} \sim \mathrm{P}(\cdot)$

1: **if** $\alpha \in \mathsf{c}$ **then return** $\mathsf{c}(\alpha)$
2: **if** $\alpha$ is a literal **then**
3:    $e \leftarrow 0$
4: **else if** $\alpha$ is an AND gate **then**
5:    $e \leftarrow \mathrm{ENT}(\beta, \mathrm{P}, \mathsf{c}) + \mathrm{ENT}(\gamma, \mathrm{P}, \mathsf{c})$
6: **else if** $\alpha$ is an OR gate **then**
7:    $e \leftarrow \sum_{i=1}^{|\mathsf{in}(\alpha)|} \mathrm{P}(\beta_i) \log \mathrm{P}(\beta_i) + \mathrm{P}(\beta_i) \mathrm{ENT}(\beta_i, \mathrm{P}, \mathsf{c})$
8: $\mathsf{c}(\alpha) \leftarrow e$
9: **return** $e$

---

## 3   COMPUTING THE LOSS

The above loss is, in general, hard to compute. To see this, consider the uniform distribution over models of a constraint $\alpha$. That is, let $\mathrm{P}(\mathbf{y}|\alpha) = \frac{1}{|m(\alpha)|}$ for all $\mathbf{y} \models \alpha$. Then, $H(\mathbf{Y}|\alpha) = -\sum_{\mathbf{y} \models \alpha} \frac{1}{|m(\alpha)|} \log \frac{1}{|m(\alpha)|} = \log |m(\alpha)|$. This tells us how many models of $\alpha$ there are, which is a well-known #P-hard problem [Valiant, 1979a,b]. We will show that, through compilation into tractable circuits, we can compute Equation 3 in time linear in the size of the circuit.

### 3.1   COMPUTATION THROUGH COMPILATION

**Tractable Circuit Compilation**   We resort to knowledge compilation techniques – a class of methods that transform, or *compile*, a logical theory into a target form with certain properties that allow certain probabilistic queries to be answered efficiently. More precisely, we know of circuit languages that compute the probability of constraints [Darwiche, 2003], and that are amenable to backpropagation. We use the circuit compilation techniques in Darwiche [2011] to build a logical circuit representing our constraint. Due to the structural properties of this circuit form, we can use it to compute both the probability of the constraint as well as its gradients with respect to the network's weights, in time linear in the size of the circuit [Darwiche and Marquis, 2002]. This does not, in general, escape the complexity of the computation: worst case, the compiled circuit can be exponential in the size of the constraint. In practice, however, constraints often exhibit enough structure (repeated sub-problems) to make compilation feasible. We refer to Section A for an illustrative example of such a compilation.

**Logical Circuits**   More formally, a *logical circuit* is a directed, acyclic computational graph representing a logical formula. Each node $n$ in the DAG encodes a logical sub-formula, denoted $[n]$. Each inner node in the graph is either an AND or an OR gate, and each leaf node encodes a Boolean literal ($Y$ or $\neg Y$). We denote by $\mathsf{in}(n)$ the set of

$n$'s children, that is, the operands of its logical gate.

**Structural Properties** As already alluded to, circuits enable the tractable computation of certain classes of queries over encoded functions granted that a set of structural properties are enforced. We explicate such properties below.

A circuit is *decomposable* if the inputs of every AND gate depend on disjoint sets of variables i.e. for $\alpha = \beta \wedge \gamma$, $\mathsf{vars}(\beta) \cap \mathsf{vars}(\gamma) = \varnothing$. Intuitively, decomposable AND nodes encode local factorizations of the function. For the sake of simplicity, we assume that decomposable AND gates always have two inputs, a condition that can be enforced on any circuit in exchange for a polynomial increase in its size [Vergari et al., 2015, Peharz et al., 2020].

A second useful property is *smoothness*. A circuit is *smooth* if the children of every OR gate depend on the same set of variables i.e. for $\alpha = \bigvee_i \beta_i$, we have that $\mathsf{vars}(\beta_i) = \mathsf{vars}(\beta_j) \, \forall i, j$. Decomposability and smoothness are a sufficient and necessary condition for tractable integration over arbitrary sets of variables in a single pass, as they allow larger integrals to decompose into smaller ones [Choi et al., 2020].

Lastly, a circuit is said to be *deterministic* if, for any input, at most one child of every OR node has a non-zero output i.e. for $\alpha = \bigvee_i \beta_i$, we have that $\beta_i \wedge \beta_j = \bot$ for all $i \neq j$. Figure 2 shows an example of smooth, decomposable and deterministic circuit.

## 3.2 ALGORITHM

Let $\alpha$ be a *smooth*, *deterministic* and *decomposable* logical circuit encoding our constraint, defined over Boolean variables $\mathbf{Y} = \{Y_1, \ldots, Y_n\}$. We now show that we can compute the constrained entropy in Equation 3 in time linear in the size of $\alpha$. The key insight is that, using circuits, we are able to efficiently decompose an expectation with respect to a fully-factorized distribution by alternately splitting the query variables and the support of the distribution until we reach the leaves of the circuit, which are simple literals. In what follows, in a slight abuse of notation for brevity, all unconditional probabilities are implicitly conditioned on constraint $\alpha$; that is we redefine $\mathrm{P}(\cdot)$ as $\mathrm{P}(\cdot|\alpha)$.

### 3.2.1 Base Case: $\alpha$ is a literal

When $\alpha$ is a literal, $\alpha = Y_i$ or $\alpha = \neg Y_i$, we have that

$$\mathrm{P}(y_i|\alpha) = \mathbb{1}\{y_i \models [\alpha]\}, \text{ and}$$
$$H(y_i|\alpha) = -\mathrm{P}(y_i|\alpha) \log \mathrm{P}(y_i|\alpha) = 0.$$

Intuitively, a literal has no uncertainty associated with it.

### 3.2.2 Recursive Case: $\alpha$ is a conjunction

When $\alpha$ is a conjunction, decomposability enables us to write

$$\mathrm{P}(\mathbf{y}|\alpha) = \mathrm{P}(\mathbf{y}_1|\beta) \, \mathrm{P}(\mathbf{y}_2|\gamma), \text{ where } \mathsf{vars}(\beta) \cap \mathsf{vars}(\gamma) = \varnothing$$

as it decomposes $\alpha$ into two independent constraints $\beta$ and $\gamma$, and $\mathbf{y}$ into two independent assignments $\mathbf{y}_1$ and $\mathbf{y}_2$. The neuro-symbolic entropy $-\mathbb{E}_{\mathbf{Y}|\alpha}[\log \mathrm{P}(\mathbf{Y}|\alpha)]$ is then

$$-\mathbb{E}_{\{\mathbf{Y}_1, \mathbf{Y}_2\}|\alpha}\Big[\log \mathrm{P}(\mathbf{Y}_1|\beta) + \log \mathrm{P}(\mathbf{Y}_2|\gamma)\Big]$$
$$= -\Big[\mathbb{E}_{\mathbf{Y}_1|\beta}\big[\log \mathrm{P}(\mathbf{Y}_1|\beta)\big] + \mathbb{E}_{\mathbf{Y}_2|\gamma}\big[\log \mathrm{P}(\mathbf{Y}_2|\gamma)\big]\Big].$$

That is, the entropy given a decomposable conjunction $\alpha$ is the sum of entropies given the conjuncts of $\alpha$.

### 3.2.3 Recursive Case: $\alpha$ is a disjunction

When $\alpha$ is a smooth and deterministic disjunction, we have that $\alpha = \bigvee_i \beta_i$, where the $\beta_i$s are mutually exclusive, and therefore partition $\alpha$. Consequently, we have that

$$\mathrm{P}(\mathbf{y}|\alpha) = \sum_i \mathrm{P}(\beta_i) \cdot \mathrm{P}(\mathbf{y}|\beta_i).$$

The neuro-symbolic entropy decomposes as well:

$$-\mathbb{E}_{\mathbf{Y}|\alpha}[\log \mathrm{P}(\mathbf{Y}|\alpha)] = -\sum_{\mathbf{y} \models \alpha} \mathrm{P}(\mathbf{y}|\alpha) \log \mathrm{P}(\mathbf{y}|\alpha)$$
$$= -\sum_{\mathbf{y} \models \alpha} \sum_i \mathrm{P}(\beta_i) \mathrm{P}(\mathbf{y}|\beta_i) \log \Big[\sum_j \mathrm{P}(\beta_j) \mathrm{P}(\mathbf{y}|\beta_j)\Big]$$
$$= -\sum_{\mathbf{y} \models \alpha} \sum_i \mathrm{P}(\beta_i) \mathrm{P}(\mathbf{y}|\beta_i) [\![\mathbf{y} \models \beta_i]\!]$$
$$\log \Big[\sum_j \mathrm{P}(\beta_j) \mathrm{P}(\mathbf{y}|\beta_j) [\![\mathbf{y} \models \beta_j]\!]\Big],$$

where by determinism, we have that, for any $\mathbf{y}$ such that $\mathbf{y} \models \alpha$, $\mathbf{y} \models \beta_i \implies \mathbf{y} \not\models \beta_j$ for all $i \neq j$. In other words, any state that satisfies the constraint $\alpha$ satisfies one and only one of its terms, and therefore, the above expression equals

$$-\sum_{\mathbf{y} \models \alpha} \sum_i \mathrm{P}(\beta_i) \mathrm{P}(\mathbf{y}|\beta_i) \log \Big[\mathrm{P}(\beta_i) \mathrm{P}(\mathbf{y}|\beta_i)\Big] [\![\mathbf{y} \models \beta_i]\!]$$
$$= -\sum_i \sum_{\mathbf{y} \models \beta_i} \mathrm{P}(\beta_i) \mathrm{P}(\mathbf{y}|\beta_i) \log \Big[\mathrm{P}(\beta_i) \mathrm{P}(\mathbf{y}|\beta_i)\Big].$$

Further simplifying the expression, expanding the logarithm, and using the fact that probability sums to 1 yields

$$= -\sum_i \mathrm{P}(\beta_i) \log \mathrm{P}(\beta_i) \sum_{\mathbf{y} \models \beta_i} \mathrm{P}(\mathbf{y}|\beta_i)$$
$$+ \mathrm{P}(\beta_i) \sum_{\mathbf{y} \models \beta_i} \mathrm{P}(\mathbf{y}|\beta_i) \log \mathrm{P}(\mathbf{y}|\beta_i)$$

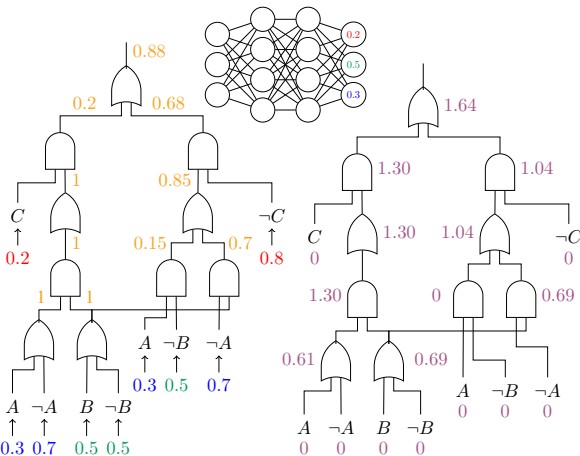

Figure 2: For a given data point, the network (middle) outputs a distribution over classes $A, B$ and $C$, highlighted in blue, green and red, respectively. The circuit encodes the constraint $(A \wedge B) \implies C$. For each leaf node $l$, we plug in $P(l)$ and $1 - P(l)$ for positive and negative literals, respectively. The computation proceeds bottom-up, taking products at AND gates and summations at OR gates. The value accumulated at the root of the circuit (left) is the probability allocated by the network to the constraint. The weights accumulated on edges from OR gates to their children are of special significance: OR nodes induce a partitioning of the distribution's support, and the weights correspond to the mass allocated by the network to each mutually-exclusive event. Complemented with a second upward pass, where the entropy of an OR node is the entropy of the distribution over its children plus the expected entropy of its children, and the entropy of an AND node is the product of its children's entropies, we get the entropy of the distribution over the constraint's models – the neuro-symbolic entropy regularization loss (right).

$$= -\sum_i P(\beta_i) \log P(\beta_i) + P(\beta_i) \mathbb{E}_{\mathbf{Y}|\beta_i} \Big[ \log P(\mathbf{Y}|\beta_i) \Big].$$

That is, the entropy of the random variable $\mathbf{Y}$ conditioned on a disjunction $\alpha$ is the sum of the entropy of the distribution induced on the children of $\alpha$, and the average entropy of its children. The full algorithm is illustrated in Algorithm 1.

## 4 AN ILLUSTRATIVE EXAMPLE

Consider Figure 2. Given a data point, the neural network defines a distribution over Boolean random variables $A, B$, and $C$, where $P(A) = \mathsf{p}_0$ and $P(\neg A) = 1 - \mathsf{p}_0$, $P(B) = \mathsf{p}_1$ and $P(\neg B) = 1 - \mathsf{p}_1$, etc. The circuit encodes the constraint $(A \wedge B) \implies C$. To compute the the probability of the constraint under the network's distribution, we feed the probabilities into the circuit, proceeding in a bottom-up fashion, taking products at AND gates and summations at OR gates, accumulating intermediate computations on the edges of

the circuit. The value accumulated at the root of the circuit is the probability mass allocated by the network to models of the formula, and corresponds to the probability of the constraint under the network's distribution – this is exactly the semantic loss, up to a negative logarithm. The weights accumulated on edges from OR gates to their children are of special significance: OR nodes induce a partitioning of the distribution's support, and the weights correspond to the mass allocated by the network to each mutually-exclusive event. Complemented with another upward pass, where the entropy of every OR node is the entropy of the distribution over it's children plus the expected entropy of its children, and the entropy of every AND node is the product of its children's entropies, we calculate the entropy of the distribution over models of the constraint – this is exactly the neuro-symbolic entropy regularization. Therefore, performing two upward sweeps of the circuit, we are able to compute the neuro-symbolic entropy regularization and the semantic loss

## 5 EXPERIMENTAL EVALUATION

In this section we set out to empirically test our neuro-symbolic entropy loss. To that end, we devise a series of semi-supervised and fully-supervised structured prediction experiments. Such are settings where, contrary to the their dominant use, classifiers are expected to predict structured objects rather than scalar, discrete or real values. Such objects are defined in terms of constraints: a set of rules characterizing the set of solutions. We aim to answer the following:

1. Does entropy regularization, in general, lead to predictive models with improved generalization capabilities?

2. If the answer to the above question is in the positive, it is our expectation that restricting the distribution acted upon by entropy regularization to that over just the models of the constraint might seem more sensible as compared to entropy-regularizing the entire predictive distribution–including non-models of the constraint. Do experiments corroborate such a hypothesis?

3. Finally, entropy regularization can be interpreted as clustering the different classes, and has intimate connections to transductive Support Vector Machines [Chapelle et al., 2010]. Does such an interpretation carry over to models and non-models of the constraint? Put differently, can we expect entropy-regularized predictive models to better conform to our constraints, measured by the percentage of predictions satisfying the constraint *regardless* of matching the groundtruth.

### 5.1 SEMI-SUPERVISED: ENTITY-RELATION EXTRACTION

We begin by testing our research questions in the semi-supervised setting. Here the model is presented with only

Table 1: Experimental results for entity-relation extraction on ACE05 and SciERC. #Labels indicates the number of labeled data points available to the network per relation. The remaining training set is stripped of labels and utilized in an unsupervised manner. We report the F1-score where a prediction is correct if the relation and its entities are correct.

| # Labels | | 3 | 5 | 10 | 15 | 25 | 50 | 75 |
|---|---|---|---|---|---|---|---|---|
| ACE05 | Baseline | $4.92 \pm 1.12$ | $7.24 \pm 1.75$ | $13.66 \pm 0.18$ | $15.07 \pm 1.79$ | $21.65 \pm 3.41$ | $28.96 \pm 0.98$ | $33.02 \pm 1.17$ |
| | Self-training | $7.72 \pm 1.21$ | $12.83 \pm 2.97$ | $16.22 \pm 3.08$ | $17.55 \pm 1.41$ | $27.00 \pm 3.66$ | $32.90 \pm 1.71$ | $37.15 \pm 1.42$ |
| | Product t-norm | $8.89 \pm 5.09$ | $14.52 \pm 2.13$ | $19.22 \pm 5.81$ | $21.80 \pm 7.67$ | $30.15 \pm 1.01$ | $34.12 \pm 2.75$ | $37.35 \pm 2.53$ |
| | Semantic Loss | $12.00 \pm 3.81$ | $14.92 \pm 3.14$ | $22.23 \pm 3.64$ | $27.35 \pm 3.10$ | $30.78 \pm 0.68$ | $36.76 \pm 1.40$ | $38.49 \pm 1.74$ |
| | + Full Entropy | $\mathbf{14.80} \pm 3.70$ | $15.78 \pm 1.90$ | $23.34 \pm 4.07$ | $28.09 \pm 1.46$ | $31.13 \pm 2.26$ | $36.05 \pm 1.00$ | $39.39 \pm 1.21$ |
| | + NeSy Entropy | $14.72 \pm 1.57$ | $\mathbf{18.38} \pm 2.50$ | $\mathbf{26.41} \pm 0.49$ | $\mathbf{31.17} \pm 1.68$ | $\mathbf{35.85} \pm 0.75$ | $\mathbf{37.62} \pm 2.17$ | $\mathbf{41.28} \pm 0.46$ |
| SciERC | Baseline | $2.71 \pm 1.10$ | $2.94 \pm 1.00$ | $3.49 \pm 1.80$ | $3.56 \pm 1.10$ | $8.83 \pm 1.00$ | $12.32 \pm 3.00$ | $12.49 \pm 2.60$ |
| | Self-training | $3.56 \pm 1.40$ | $3.04 \pm 0.90$ | $4.14 \pm 2.60$ | $3.73 \pm 1.10$ | $9.44 \pm 3.80$ | $14.82 \pm 1.20$ | $13.79 \pm 3.90$ |
| | Product t-norm | $\mathbf{6.50} \pm 2.00$ | $8.86 \pm 1.20$ | $10.92 \pm 1.60$ | $13.38 \pm 0.70$ | $13.83 \pm 2.90$ | $19.20 \pm 1.70$ | $19.54 \pm 1.70$ |
| | Semantic Loss | $6.47 \pm 1.02$ | $\mathbf{9.31} \pm 0.76$ | $11.50 \pm 1.53$ | $12.97 \pm 2.86$ | $14.07 \pm 2.33$ | $20.47 \pm 2.50$ | $23.72 \pm 0.38$ |
| | + Full Entropy | $6.26 \pm 1.21$ | $8.49 \pm 0.85$ | $11.12 \pm 1.22$ | $14.10 \pm 2.79$ | $17.25 \pm 2.75$ | $\mathbf{22.42} \pm 0.43$ | $24.37 \pm 1.62$ |
| | + NeSy Entropy | $6.19 \pm 2.40$ | $8.11 \pm 3.66$ | $\mathbf{13.17} \pm 1.08$ | $\mathbf{15.47} \pm 2.19$ | $\mathbf{17.45} \pm 1.52$ | $22.14 \pm 1.46$ | $\mathbf{25.11} \pm 1.03$ |

a portion of the labeled training set, with the rest used exclusively in an unsupervised manner by the respective approaches.

We make use of the natural ontology of entity types and their relations present when dealing with relational data. This defines a set of relations and their permissible argument types. As is with all of our constraints, we express the aforementioned ontology in the language of Boolean logic.

Our approach to recognizing the named entities and their pairwise relations is most similar to Zhong and Chen [2020]. Contextual embeddings are first procured for every token in the sentence. These are then fed into a named entity recognition module that outputs a vector of per-class probability for every entity. A classifier then classifies the concatenated contextual embeddings and entity predictions into a relation.

We employ two entity-relation extraction datasets, the Automatic Content Extraction (ACE) 2005 [Walker et al., 2006] and SciERC datasets [Luan et al., 2018]. ACE05 defines an ontology over 7 entities and 18 relations from mixed-genre text, whereas SciERC defines 6 entity types with 7 possible relation between them and includes annotations for scientific entities and there relations, assimilated from 12 AI conference/workshop proceedings. We report the percentage of coherent predictions: data points for which the predicted entity types, as well as the relations are correct.

We compare against five baselines. The first baseline is a purely supervised model which makes no use of unlabeled data. The second is a classical self-training approach based off of Chang et al. [2007], and uses integer linear programming to impute the unlabeled data's most likely labels subject to the constraint, and consequently augment the (small) labeled set. The third baseline is a popular instantiation of a broad class of methods, fuzzy logics, which replace logical operators with their fuzzy t-norms and logical implications

Table 2: Grid shortest path test results

| Test accuracy % | Coherent | Incoherent | Constraint |
|---|---|---|---|
| 5-layer MLP | 5.62 | **85.91** | 6.99 |
| Semantic loss | 28.51 | 83.14 | 69.89 |
| + Full Entropy | 29.02 | 83.76 | 75.23 |
| + NeSy Entropy | **30.12** | 83.01 | **91.61** |

with simple inequalities. Lastly, we compare our proposed method, dubbed "NeSy Entropy", to vanilla semantic loss as proposed in Xu et al. [2018] as well as another entropy-regularized baseline, dubbed "Full Entropy", which minimizes the entropy of the entire predictive distribution, as opposed to just the distribution over the constraint's models.

Our results are shown in Table 1. We observe that semantic loss outperforms the baseline, self-training, and product t-norm across the board. We attribute such a performance to the exactness of semantic loss, and its faithfulness to the underlying constraint. We also observe that entropy-regularizing the predictive model, in conjunction with training using semantic loss leads to better predictive models, as compared with models trained solely using semantic loss. Furthermore, it turns out that restricting entropy to the distribution over the constraint's models, models that we know constitute the set of valid predictions, compared to the model's entire predictive distribution, which includes valid and invalid predictions, leads to a non-trivial increase in the accuracy of predictions.

## 5.2 FULLY-SUPERVISED LEARNING

We now turn our attention to testing our hypotheses in a fully supervised setting, where our aim is to examine the effect of constraints enforced on the training set. We note that this is

Table 3: Preference prediction test results

| Test accuracy % | Coherent | Incoherent | Constraint |
|---|---|---|---|
| 3-layer MLP | 1.01 | **75.78** | 2.72 |
| Semantic loss | 15.03 | 72.43 | 69.83 |
| + Full Entropy | 17.52 | 71.80 | 80.21 |
| + NeSy Entropy | **18.17** | 71.51 | **96.04** |

Table 4: Warcraft shortest path prediction results

| Test accuracy % | Coherent | Incoherent | Constraint |
|---|---|---|---|
| ResNet-18 | 44.8 | 97.7 | 56.9 |
| Semantic loss | 50.9 | 97.7 | 67.4 |
| + Full Entropy | 51.5 | 97.6 | 67.7 |
| + NeSy Entropy | **55.0** | **97.9** | **69.8** |

a seemingly harder setting in the following sense: In a semi-supervised setting we might make the argument that, despite its abundance, imposing an auxiliary loss on unlabeled data provides the predictive model with an unfair advantage as compared to the baseline. We concern ourselves with two tasks: predicting paths in a grid and preference learning.

**Predicting Simple Paths** For this task, our aim is to find the shortest path in a graph, or more specifically a 4-by-4 grid, $G = (V, E)$ with uniform edge weights. Our input is a binary vector of length $|V| + |E|$, with the first $|V|$ variables indicating the source and destination, and the next $|E|$ variables encoding a subgraph $G' \subseteq G$. Each label is a binary vector of length $|E|$ encoding the shortest *simple* path in $G'$, a requirement that we enforce through our constraint. We follow the algorithm proposed by Nishino et al. [2017] to generate a constraint for each simple path in the grid, conjoined with indicators specifying the corresponding source-destination pair. Our constraint is then the disjunction of all such conjunctions.

To generate the data, we begin by randomly removing one third of the edges in the graph $G$, resulting in a subgraph, $G'$. Subsequently, we filter out connected components in $G'$ with fewer than 5 nodes to reduce degenerate cases. We then sample a source and destination node uniformly at random. The latter constitutes a single data point. We generate a dataset of 1600 examples, with a $60/20/20$ train/validation/test split.

**Preference Learning** We also consider the task of preference learning. Given the user's ranking of a subset of elements, we wish to predict the user's preferences over the remaining elements of the set. We encode an ordering over $n$ items as a binary matrix $X_{ij}$, where for each $i, j \in 1, \ldots, n$, $X_{ij}$ denotes that item $i$ is at position $j$. Our constraint $\alpha$ requires that the network's output be a valid total ordering. We use preference ranking data over 10 types of sushi for

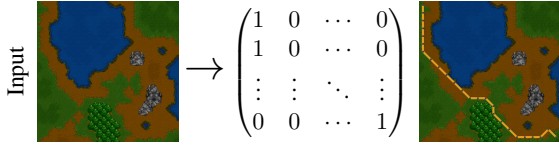

Figure 3: Warcraft dataset. Each input (left) is a $12 \times 12$ grid corresponding to a Warcraft II terrain map, the output is a matrix (middle) indicating the shortest path from top left to bottom right (right).

$5,000$ individuals, taken from PREFLIB [Mattei and Walsh, 2013], split 60/20/20. Our inputs consist of the user's preference over 6 sushi types, with the model tasked to predict the user's preference, a *strict* total order, over the remaining 4.

Tables 2 and 3 compares the baseline to the same MLP augmented with semantic loss, semantic loss with entropy regularization over the entire predictive distribution, dubbed "Full Entropy" and entropy regularization over the distribution over the constraint's models, dubbed "NeSy Entropy".

Similar to Xu et al. [2018], we observe that the semantic loss has a marginal effect on incoherent accuracy, but significantly improves the network's ability to output coherent predictions. We also observe that, similar to semi-supervised settings, entropy-regularization leads to more coherent predictions using both "Full Entropy" and "NeSy Entropy", with "NeSy Entropy" leading to the best performing predictive models. Remarkably, we also observe that "NeSy Entropy" leads to predictive models whose predictions almost always satisfy the constraint, captured by "Constraint".

**Warcraft Shortest Path** Lastly, we consider a more real-world variant of the task of predicting simple paths. Following [Pogančić et al., 2020], our training set consists of $10,000$ terrain maps curated using Warcraft II tileset. Each map encodes an underlying grid of dimension $12 \times 12$, where each vertex is assigned a cost depending on the type of terrain it represents (e.g. earth has lower cost than water). The shortest (minimum cost) path between the top left and bottom right vertices is encoded as an indicator matrix, and serves as label. Figure 3 shows an example input presented to the network, the groundtruth, and the input with the annotated shortest path. Figure 4 shows examples of baseline predictions and those obtained by training with constraints.

Presented with an image of a terrain map, a convolutional neural network – following [Pogančić et al., 2020], we use ResNet18 [He et al., 2016] – outputs a $12 \times 12$ binary matrix indicating the vertices that constitute the minimum cost path. We report three metrics: "Coherent" denotes the percentage of optimal-cost predictions, "Incoherent" denotes the percentage of individual vertices matching the groundtruth, and "Constraint" indicates the percentage of predictions that constitute valid paths. Our results are shown in Table 4.

In line with our previous experiments, we observe that incor-

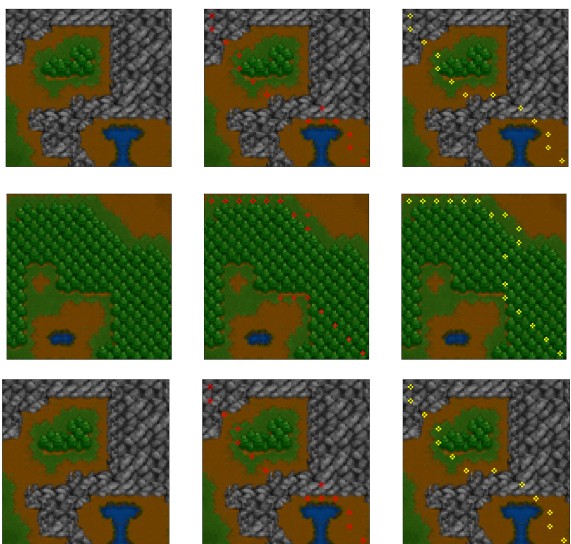

Figure 4: Example maps from the Warcraft dataset (left) annotated with the baseline predictions in red (center), and the predictions obtained using constraints in yellow (right)

porating constraints into learning improves the "Coherent" metric from 44.8% to 50.9%, and of the "Coherent" metric from 56.9% to 67.4%. Augmenting semantic loss with the entropy over the network's predictive distribution, "Full Entropy", we attain a modest improvement from 50.9% to 51.5% and 67.4% to 67.7% for the "Coherent" and "Constraint" metrics respectively. Restricting the entropy minimization to models of the constraint, "NeSy Entropy", we observe that we attain a large improvement to 55.0% and 69.8% for the "Coherent" and "Constraint" metrics resp.

# 6 RELATED WORK

The idea of using a model's predictions to obtain artificial labels for unlabeled data is as old as time [Scudder, 1965, McLachlan, 1975], and has often known throughout the literature as pseudo-labeling or self-training. Self-training is an iterative process by which a learner imputes the labels of examples which have been confidently classified in the previous step, and can therefore be viewed as implicitly minimizing the model's entropy. This is done explicitly in Grandvalet and Bengio [2005] with a loss term which minimizes the entropy of the model's predicted distribution for any given unlabeled data point, thereby rendering the entropy computation amenable to differentiation, and allowing finer control on the influence of the unlabeled data. It has been applied successfully across a wide range of domain, including NLP [McClosky et al., 2006], object detection [Rosenberg et al., 2005], image classification [Lee, 2013, Xie et al., 2019], domain adaptation [Zou et al., 2018], to name a few. It has also been used recently by a plethora of semi-supervised learning algorithms as a constituent of their

training pipelines [Arazo et al., 2019, Pham and Le, 2019, Miyato et al., 2018, Berthelot et al., 2019]. This is in contrast to entropy maximization, used in reinforcement learning, where the aim is to capture the entire range of low-cost behaviors, not a single correct one[Toussaint, 2009].

In an acknowledgment to the need for both symbolic as well as sub-symbolic reasoning, there has been a plethora of recent works studying how to best combine neural networks and logical reasoning, dubbed *neuro-symbolic reasoning*. The focus of such approaches is typically making probabilistic reasoning tractable through first-order approximations, and differentiable, through reducing logical formulas into arithmetic objectives, replacing logical operators with their fuzzy t-norms, and implications with inequalities [Kimmig et al., 2012, Rocktäschel et al., 2015, Fischer et al., 2019].

Constraint driven learning [Chang et al., 2007] is a classic work that lies at the intersection of both bodies of work. Therein, in a fashion similar to self-training, the learner imputes the labels of the samples that were confidently classified *subject to the constraint*. Therefore, the imputed labels are guaranteed to be valid. CoDL, however, performs a first-order approximation, approximating the netwok's full posterior by the MAP. Furthermore it is not differentiable.

Diligenti et al. [2017] and Donadello et al. [2017] use first-order logic to specify constraints on outputs of a neural network. They employ fuzzy logic to reduce logical formulas into differential, arithmetic objectives denoting the extent to which neural network outputs violate the constraints, thereby supporting end-to-end learning under constraints. More recently, Xu et al. [2018] introduced semantic loss, which circumvents the shortcomings of fuzzy approaches, while still supporting end-to-end learning under constraints. More precisely, *fuzzy reasoning* is replaced with *exact probabilistic reasoning*, made possible by compiling logical formulae into structures supporting efficient probabilistic queries.

Another class of neuro-symbolic approaches have their roots in logic programming. DeepProbLog [Manhaeve et al., 2018] extends ProbLog, a probabilistic logic programming language, with the capacity to process neural predicates, whereby the network's outputs are construed as the probabilities of the corresponding predicates. This simple idea retains all essential components of ProbLog: the semantics, inference mechanism, and the implementation. In a similar vein, Dai et al. [2018] combine domain knowledge specified as purely logical Prolog rules with the output of neural networks, dealing with the network's uncertainty through revising the hypothesis by iteratively replacing the output of the neural network with anonymous variables until a consistent hypothesis can be formed. Bošnjak et al. [2017] present a framework combining prior procedural knowledge, as a Forth program, with neural functions learned through data. The resulting neural programs are consistent with specified prior knowledge and optimized with respect to data.

# 7 CONCLUSION

In conclusion, we proposed neuro-symbolic entropy regularization, a principled approach to unifying neuro-symbolic learning and entropy regularization. It encourages the network to output distributions that are peaked over models of the logical formula. We are able to compute our loss due to structural properties of circuit languages. We validate our hypothesis on four different tasks under semi-supervised and fully-supervised settings and observed an increase in *accuracy* as well as the *validity* of the model's predictions.

### Acknowledgements

KA would like to thank Arthur Choi, Antonio Vergari, Yoojung Choi, and Tal Friedman for helpful discussions throughout the project. This work is partially supported by a DARPA PTG grant, NSF grants #IIS-1943641, #IIS-1956441, #CCF-1837129, Samsung, CISCO, and a Sloan Fellowship.

# A COMPILING LOGICAL FORMULAS INTO TRACTABLE CIRCUITS

At a high level, there exist off-the-shelf compilers [Choi and Darwiche, 2013, Oztok and Darwiche, 2015, Darwiche, 2004, Muise et al., 2012, Lagniez and Marquis, 2017, Toda and Soh, 2016] utilizing SAT solvers, essentially through case analysis, to compile a logical formula into a tractable logical circuit. NeSy Entropy is agnostic to the exact flavor of circuit so long as the properties outlined in Section 3.2 are respected. In our experiments, we use PySDD[1] a python SDD compiler [Darwiche, 2011, Choi and Darwiche, 2013]. We will now step through an example of compiling a logical formula. Consider the circuit in Figure 2 encoding constraint

$$(A \wedge B) \implies C,$$

to be construed as encoding, animal $\wedge$ barks $\implies$ dog.

Intuitively, our aim is to transform the above logical formula into a *compact* target form representing all possible assignments to $A, B$ and $C$ satisfying the logical formula. We compile such a constraint by proceeding in a bottom up fashion, where bottom-up compilation can be seen as composing Boolean sub-functions whose domain is determined by a variable ordering. Concretely, starting from circuits for literals $A$ and $B$, we compile a circuit $\beta = A \wedge B$. We compose the previously compiled circuit $\beta$ with the circuit for literal $C$. We point out that this is achieved using a couple of simple API calls to a bottom-up compiler. We will now step through the actual construction of the circuit. We introduce logical circuits representing the literals

$$A \qquad \neg A \qquad B \qquad \neg B \qquad C \qquad \neg C$$

[1]https://github.com/wannesm/PySDD

The compiler disjoins literals $A$ with $\neg A$, and $B$ with $\neg B$, introducing deterministic and smooth OR nodes.

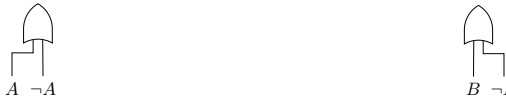

An OR node represents *disjoint solutions* to the logical formula, meaning there exists distinct assignments, characterized by the children, satisfying the constraint e.g. $a, \neg a, b$ and $\neg b$ all occur as part of distinct solutions to the constraint.

Compilation proceeds by conjoining constraint circuits for $A \vee \neg A$ with $B \vee \neg B$, $\neg A$ with $B \vee \neg B$ and $A$ with $\neg B$.

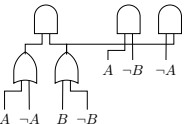

Decomposable AND nodes *compose* functions over *disjoint sets of variables*. These AND nodes represent Boolean functions $(A \vee \neg A) \wedge (B \vee \neg B)$, $\neg A \wedge (B \vee \neg B)$, and $A \wedge \neg B$.

The compiler disjoins $\neg A \wedge (B \vee \neg B)$, with $A \wedge \neg B$ and $(A \vee \neg A) \wedge (B \vee \neg B)$ with true, the multiplicative identity, guaranteeing alternating AND and OR nodes, for convenience. It is worth reiterating that every child of an OR node encodes disjoint solutions over the same set of variables.

So far, we have compiled logical circuits for the formula

$$(\neg A \wedge (B \vee \neg B)) \vee (A \wedge \neg B) \qquad (4)$$

as well as for the fomula

$$(A \vee \neg A) \wedge (B \vee \neg B) \qquad (5)$$

What remains is to conjoin eq. (4) with $C$, and eq. (5) with $\neg C$, and disjoin the resulting circuits. What we get is a disjunction over the possible solutions of the constraint: predicting the presence of a barking animal implies the presence of a dog. Otherwise, there might or not be a dog.

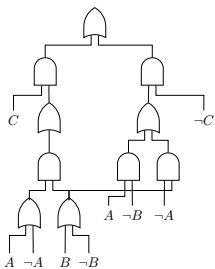

Compilation techniques like the one we illustrated do not, however, escape the hardness of the problem: the compiled circuit can be exponential in the size of the constraint, *in the worst case*. *In practice*, however, we can obtain compact circuits because real-life logical constraints exhibit enough structure (e.g., repeated sub-problems) that can be easily exploited by a compiler [Darwiche and Marquis, 2002].

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
