# OpenReview forum: "Neuro-Symbolic Entropy Regularization"
_auai.org/UAI/2022/Conference — UAI 2022 Oral_

### Official Review · Reviewer_yJcU · 2022-03-18

**Q2(1) Originality/Novelty:** 4
**Q2(2) Significance/Impact:** 4
**Q2(3) Correctness/Technical Quality:** 4
**Q2(6) Clarity Of Writing:** 4
**Q6 Overall Score:** 9
**Q8 Confidence In Your Score:** 5

**Q1 Summary And Contributions:**

The paper combines semantic loss with entropy regularization, obtaining a neuro-symbolic form of entropy regularization.
The paper proposes a form of tractable circuits to represent the constraints on the output and computes the entropy
of the output.
The approach is experimented on entity-relation extraction in a semi-supervised fashion and in predicting paths in graphs
and in Warcraft maps in a fully-supervised fashion. The approach significantly improves over various baselines.


**Q2 Assessment Of The Paper:**

More detailed information regarding each of these aspects is given below:

**Q2(4) Quality Of Experiments (Optional):**

4: Excellent: The experimental evaluation is comprehensive and the results are compelling.

**Q2(5) Reproducibility:**

2: Fair: Key resources (e.g., proofs, code, data) are unavailable but key details (e.g., proof sketches, experimental setup) are sufficiently well-described for an expert to confidently reproduce the main results.

**Q3 Main Strengths:**

The idea of combining semantic loss with entropy regularization is novel and very interesting.
The use of knowledge compilation is a clever way of ensuring that the computation of the entropy is fast.
This approach will surely have a significant impact on the wide subfield of AI concerned with structured prediction and
on the subfield of neuro-symbolic integration.
Technically the paper appears sound.
The experiments are extensive and convincing, touching both semi-supervised and fully supervised learning.
The baselines are sufficiently representative of the state of the art and the improvements are significative.
Apart from the fact that the paper does not contain links to code and data, the results seems reproducible.
The paper is overall very clearly written.

**Q4 Main Weakness:**

Links to code and data are missing

**Q5 Detailed Comments To The Authors:**

In Section 3.2.3 the computation of Pr(y|\alpha) seems to miss a 1/Pr(\alpha) factor. Since this is a constant it does not influence the loss but the formula as it is is incorrect.

On page 2 the sentence "The notation
for states y is used to refer to an assignment, the logical
sentence enforcing the assignment, or the binary output
vector capturing the assignment, as these are all equivalent
notions." is unclear: in which sense y refers to the logical sentence enforcing the assignment?

The sentence "Entropy-regularization [Grandvalet
and Bengio, 2005] directly implements the above assumption,
requiring the classifier output confident – low-entropy
– predictive distributions" is difficult to parse grammatically.


Table 3 and Figure 4 are not referenced in the text

The reference "Wang-Zhou Dai, Qiu-Ling Xu, Yang Yu, and Zhi-Hua Zhou.
Tunneling neural perception and logic reasoning through
abductive learning, 2018." misses the venue.

**Q7 Justification For Your Score:**

I liked very much the idea of combining semantic loss with entropy regularization and the use of knowledge compilation to
ensure a fast computation of the loss.
The paper clearly explains the ideas and presents convincing experimental evidence.

**Q9 Complying With Reviewing Instructions:**

1: Yes.

---

### Official Review · Reviewer_vCAu · 2022-04-10

**Q2(1) Originality/Novelty:** 2
**Q2(2) Significance/Impact:** 2
**Q2(3) Correctness/Technical Quality:** 3
**Q2(6) Clarity Of Writing:** 4
**Q6 Overall Score:** 5
**Q8 Confidence In Your Score:** 4

**Q1 Summary And Contributions:**

The paper proposes a novel loss function that unifies neuro-symbolic computations with entropy regularization and is especially useful in multi output variable prediction i.e. structured prediction. The loss is closely related to the semantic loss and is compiled into a circuit, to take advantage of their tractability property, in order to be computed efficiently. Thorough experimental section shows the effectiveness of the proposed loss function in various tasks.

**Q2 Assessment Of The Paper:**

More detailed information regarding each of these aspects is given below:

**Q2(4) Quality Of Experiments (Optional):**

3: Good: The experimental evaluation is adequate, and the results convincingly support the main claims.

**Q2(5) Reproducibility:**

2: Fair: Key resources (e.g., proofs, code, data) are unavailable but key details (e.g., proof sketches, experimental setup) are sufficiently well-described for an expert to confidently reproduce the main results.

**Q3 Main Strengths:**

1. The paper tackles an important problem of taking structure of the output space into account during entropy regularization. The neuro-symbolic loss function is a novel attempt in this direction as far as I am aware.

2. The paper is very well written and is easy to read.

3. The experimental section is extensive and the performance of the loss function is also quite reasonable.


**Q4 Main Weakness:**

1. I think the knowledge compilation part is a little bit rushed and should be explained in a little more detail. instead of leaving it to literature

2. Also the efficiency of the compilation into  a logistic circuit should be discussed as well as a citation of "Learning Logistic Circuits, Liang and Broeck, AAAI 2019" should be added.

3. How do this loss function relate to say, label smoothing that is used in entropy regularization? Is it the addition of desired constraints as opposed to undesired constraints in label smoothing the difference? I would like to see this discussed.

**Q5 Detailed Comments To The Authors:**

I think this is a good paper with an exciting contribution of combining neuro-symbolic AI with entropy regularization, In addition to my comments in Q4  I have a few pointers/questions here:

1.  As the authors themselves mention, the compiled circuit can be exponential in the size of the constraint in the worst case. How can this be alleviated since scaling up the data sets will lead to this problem. I also partially disagree with the fact that in practice this is taken care of by the constraints automatically. It might be true for simple data sets but certainly not true for most real-world applications.

2. How will this loss function be effected by increase in number of constraints? Also how difficult will it be to design contraints for complex data sets?

3. Can these constraints be learned? That would be more interesting problem.

4. Also, I missed if there was any future work mentioned by the authors. Is there no plan of extending this work?

Overall, a nice paper but with a few issues unanswered. I would be willing to reconsider my rating during the rebuttal.

**Q7 Justification For Your Score:**

I read the paper completely and overall like the paper but have my concerns as raised in Q4 and 5.

**Q9 Complying With Reviewing Instructions:**

1: Yes.

---

### Official Review · Reviewer_NMSw · 2022-04-11

**Q2(1) Originality/Novelty:** 3
**Q2(2) Significance/Impact:** 3
**Q2(3) Correctness/Technical Quality:** 3
**Q2(6) Clarity Of Writing:** 3
**Q6 Overall Score:** 7
**Q8 Confidence In Your Score:** 2

**Q1 Summary And Contributions:**

The authors are concerned with neuro-symbolic learning / reasoning and propose a entropy regularization loss that unifies neuro-symbolic learning and entropy regularization. They show that this causes a network to produce output distributions in line with logical formulas. They show how to compute the loss using circuit languages. Experiments demonstrate validate the approach and show it to work better than previous baselines.

**Q2 Assessment Of The Paper:**

More detailed information regarding each of these aspects is given below:

**Q2(4) Quality Of Experiments (Optional):**

3: Good: The experimental evaluation is adequate, and the results convincingly support the main claims.

**Q2(5) Reproducibility:**

3: Good: Key resources (e.g., proofs, code, data) are available and key details (e.g., proofs, experimental setup) are sufficiently well-described for competent researchers to confidently reproduce the main results.

**Q3 Main Strengths:**

The paper addresses an important emerging topic (neuro-symbolic learning) and proposes an apparently novel approach for realizing this.
The paper is well written, nicely illustrated, and its content appears to be sound and solid. Experiments appear to be reasonable and carefully executed. Experimental results corroborate the validity and utility of the proposed method.

**Q4 Main Weakness:**

To readers not very familiar with the required background (including this reviewer), the content of section 3 is a bit rough. I.e. it is hard to follow and has to be taken at face value.

**Q5 Detailed Comments To The Authors:**

Though space is scarce and page restrictions are limiting, it might be a good idea to provide readers with a somewhat more extended introduction into "computation through compilation".

**Q7 Justification For Your Score:**

The paper deals with a timely and important topic, namely techniques for integrating structured knowledge into neural learning system. Although I may not have gotten every minute detail, it appears that the proposed approach is novel and original and that it seems to work well. Overall, I;d think this paper has the potential to trigger further research in this direction and thus argue for its acceptance.

**Q9 Complying With Reviewing Instructions:**

1: Yes.

---

### Decision · Program_Chairs · 2022-05-15

**Decision:**

Accept (Oral)

**Comment:**

Meta Review: The paper proposes  an entropy regularization for structured neural network outputs. The output structure is provided via a knowledge-compilation approach by compiling logical constraints into a circuit representation which then serves as a regularization term for the neural network.

Pros:
sound solution to an important and timely problem (neuro-symbolic integration)

Cons:
background (knowledge compilation) not adequately introduced for newcomers;
knowledge compilation puts certain constraints on the approach (scalability)

Quality: Good

Clarity: Fair-Good

Originality: Good

Significance: Very Good